# Improvements in Maturity and Stability of 3D iPSC-Derived Hepatocyte-like Cell Cultures

**DOI:** 10.3390/cells12192368

**Published:** 2023-09-27

**Authors:** Siiri Suominen, Tinja Hyypijev, Mari Venäläinen, Alma Yrjänäinen, Hanna Vuorenpää, Mari Lehti-Polojärvi, Mikko Räsänen, Aku Seppänen, Jari Hyttinen, Susanna Miettinen, Katriina Aalto-Setälä, Leena E. Viiri

**Affiliations:** 1Heart Group, Finnish Cardiovascular Research Center and Science Mimicking Life Research Center, Faculty of Medicine and Health Technology, Tampere University, 33520 Tampere, Finlandleena.viiri@tuni.fi (L.E.V.); 2Adult Stem Cell Group, Faculty of Medicine and Health Technology, Tampere University, 33520 Tampere, Finland; 3Research, Development and Innovation Centre, Tampere University Hospital, 33520 Tampere, Finland; 4Computational Biophysics and Imaging Group, Faculty of Medicine and Health Technology, Tampere University, 33520 Tampere, Finland; 5Department of Technical Physics, University of Eastern Finland, 70210 Kuopio, Finland; 6Heart Hospital, Tampere University Hospital, 33520 Tampere, Finland

**Keywords:** induced pluripotent stem cells (iPSCs), hepatocyte-like cell (HLC), 3D liver modeling, spheroid culturing, microfluidic chip, biomaterial, optical projection tomography (OPT), electrical impedance tomography (EIT)

## Abstract

Induced pluripotent stem cell (iPSC) technology enables differentiation of human hepatocytes or hepatocyte-like cells (iPSC-HLCs). Advances in 3D culturing platforms enable the development of more in vivo-like liver models that recapitulate the complex liver architecture and functionality better than traditional 2D monocultures. Moreover, within the liver, non-parenchymal cells (NPCs) are critically involved in the regulation and maintenance of hepatocyte metabolic function. Thus, models combining 3D culture and co-culturing of various cell types potentially create more functional in vitro liver models than 2D monocultures. Here, we report the establishment of 3D cultures of iPSC-HLCs alone and in co-culture with human umbilical vein endothelial cells (HUVECs) and adipose tissue-derived mesenchymal stem/stromal cells (hASCs). The 3D cultures were performed as spheroids or on microfluidic chips utilizing various biomaterials. Our results show that both 3D spheroid and on-chip culture enhance the expression of mature liver marker genes and proteins compared to 2D. Among the spheroid models, we saw the best functionality in iPSC-HLC monoculture spheroids. On the contrary, in the chip system, the multilineage model outperformed the monoculture chip model. Additionally, the optical projection tomography (OPT) and electrical impedance tomography (EIT) system revealed changes in spheroid size and electrical conductivity during spheroid culture, suggesting changes in cell–cell connections. Altogether, the present study demonstrates that iPSC-HLCs can successfully be cultured in 3D as spheroids and on microfluidic chips, and co-culturing iPSC-HLCs with NPCs enhances their functionality. These 3D in vitro liver systems are promising human-derived platforms usable in various liver-related studies, specifically when using patient-specific iPSCs.

## 1. Introduction

Developing an ideal in vitro liver model is mainly dependent on mimicking the key elements of the biological properties and the physiological environment of the liver [1]. Ideally, the model will reflect the in vivo-like multi-cell type environment necessary for liver responses. The liver is a complex organ composed of many cell types, but many in vitro studies are performed in 2D cultures that do not accurately reflect the complex nature of the liver architecture and function in vivo. Moreover, even though the liver tissue is composed of several cell types, including hepatocytes, and nonparenchymal cells (NPC) such as cholangiocytes, hepatic stellate, sinusoidal, and Kupffer cells as well as vascular structures, in cell culture experiments, hepatocyte monocultures are typically used. This results in lack of the structural, biochemical, and functional characteristics of the liver tissue [2].

Novel human in vitro 3D liver models offer an alternative to animal models for drug development, toxicity studies, and disease modeling. The 3D models provide structural organization with more cell–cell and cell–extracellular matrix (ECM) contacts, thus promoting liver functions. Various 3D culture systems can be utilized, from bio-printed structures to microfluidic chips or spheroid culturing, either as monocultures of hepatocytes or co-cultures with NPCs [3,4,5]. Primary human hepatocytes (PHHs) represent the current gold standard cell type in liver modeling, but they have a tendency to rapidly lose their functionality in vitro [6,7]. Induced pluripotent stem cell-derived hepatocyte-like cells (iPSC-HLCs) are a promising alternative cell source for the in vitro 3D liver models due to specific features such as their patient specificity and unlimited supply. The iPSC-HLCs, however, are still at an immature state expressing fetal markers such as alpha-fetoprotein (AFP), so their phenotypic maturity needs to be increased by better mimicking the in vivo environment through 3D culturing as well as co-culturing with other relevant cell types as well as introducing vasculature to the model [8,9].

Since hepatocytes and endothelial cells (ECs) constitute the majority of the cell population in the liver, it is essential to introduce ECs into in vitro liver models. Moreover, it was suggested that adding human adipose tissue-derived mesenchymal stem/stromal cells (hASCs) to the co-culture system could promote vascularization since these cells express several important angiogenic growth factors [10,11,12]. Complex co-culture systems also demand optimization of the cell culture media to support the hepatic differentiation [13] as well as choosing the optimal cell culture system, specifically when developing 3D in vitro models.

In this work, we explored whether 3D spheroids and microfluidic chips containing iPSC-HLCs alone or together with NPCs would recapitulate the human liver architecture and physiology better than 2D monocultures of hepatocytes. We also introduce our OPT-EIT imaging technique for monitoring the spheroids during culture. Our 3D iPSC-HLC liver models offer valuable tools for further studies exploring, e.g., the lipid metabolism, various phases of the atherosclerosis pathology or drug metabolism in a patient-specific manner, or for representing liver in any multi-organ models.

## 2. Materials and Methods

### 2.1. Induced Pluripotent Stem Cell (iPSC) Reprogramming and Cell Culture

The iPSC lines were produced from skin fibroblasts with Sendai virus vectors (OCT4, SOX2, KLF4, C-MYC; CytoTune; Life Technologies, Carlsbad, CA, USA) and maintained as published before [14]. Two patient-derived iPSC lines (UTA.10211.EURCAs and UTA.11304.EURCCs) were used in this study, and the lines were characterized as previously described [15]. Written and informed consent was obtained from all study subjects. The Ethics Committee of Tampere University Hospital approved the study and patient recruitment (approval number: R12123) and all experiments were performed in accordance with relevant guidelines and regulations.

### 2.2. Differentiation of iPSCs into Hepatocyte-like Cells (HLCs)

We used a previously published method to differentiate iPSCs to HLCs slightly modified from the method of Kajiwara et al. [16], consisting of three main stages [17]. In short, when 70% confluent, iPSCs were detached by Versene (Gibco, Thermo Fisher Scientific, Waltham, MA, USA) and re-suspended in RPMI+Glutamax (Gibco) supplemented with 1×B27 (Gibco), 100 ng/mL activin A (PeproTech, Cranbury, NJ, USA), 50 ng/mL Wnt3 (R&D Systems, Minneapolis, MN, USA), and 10 µM Rock inhibitor (STEMCELL Technologies, Vancouver, BC, Canada) and seeded with 5–10 × 10^4^ /cm^2^ density. On the next day, Rock inhibitor was replaced with 0.5 µM NaB (Sigma-Aldrich, Saint Louis, MO, USA) until day 5–6 of differentiation. In some experiments, the STEMdiff DE kit (STEMCELL Technologies) was used at this stage according to the manufacturer’s protocol. In the second phase, the hepatic differentiation was initiated by switching the medium to KO-DMEM+20% KO-SR (Gibco), 1 mM Glutamax (Gibco), 1% NEAA (Lonza, Basel, Switzerland), 0.1% β-ME (Lonza), and 1% DMSO (Sigma-Aldrich) for 7 days. From this stage onward until the end of each experiment, cells were cultured in HBM (cc-3199, Lonza) supplemented with single quotes complemented with 25 ng/mL HGF (PHG0254, Life Technologies) and 20 ng/mL oncostatin M (OSM, 295-OM, R&D Systems).

### 2.3. Isolation and Culture of Human Adipose Tissue-Derived Stem/Stromal Cells

Human adipose tissue-derived mesenchymal stem/stromal cells (hASCs) were isolated from subcutaneous abdominal tissue samples from three donors. Tissue samples were obtained at the Tampere University Hospital Department of Plastic Surgery with the donor’s written informed consent and processed under ethical approval of the Ethics Committee of the Expert Responsibility area of Tampere University Hospital (R15161), and all experiments were performed in accordance with relevant guidelines and regulations. The cells were isolated as described previously [18]. The hASCs were cultured in α-MEM supplemented with 5% human serum (HS; Biowest, Nuaillé, France; or Serana, Brandenburg, Germany), 100 U/mL penicillin, and 100 µg/mL streptomycin (Pen/Strep; Lonza) and used in a microvascular network model in passages 2–5. The mesenchymal origin of hASCs was confirmed with surface marker expression analysis with flow cytometry and assessment of adipogenic and osteogenic differentiation potential of the cells as described previously [12].

### 2.4. Isolation and Culture of Human Umbilical Vein Endothelial Cells (HUVECs)

Pooled human umbilical vein endothelial cells (HUVEC) expressing green fluorescent protein (GFP) were commercially obtained from Cellworks. GFP-HUVECs were cultured in the Endothelial Cell Growth Medium-2 Bullet Kit (EGM-2; Lonza). Instead of the fetal bovine serum supplied with the kit, 2% HS (Biowest or Serana) was used. In the microvascular network model, GFP-HUVECs were used in passages 4–6.

HUVECs isolated from human umbilical cord vein tissue samples were received from two donors. Tissue samples were obtained in planned cesarean sections at the Tampere University Hospital with the patient’s written informed consent together with the favorable opinion of the Regional Ethics Committee of the Expert Responsibility area of Tampere University Hospital, Tampere, Finland (R13019). Cells were isolated using the enzymatic procedure as previously described [19,20] apart from using the collagenase type II (1.0 mg/mL; Merck Millipore, Burlington, MA, ÙSA) enzyme in cell isolation. HUVECs were cultured in EGM-2 medium, where fetal bovine serum was replaced with 2% HS (Biowest). The cells were used in a microvascular network model in passage 4 and multilineage spheroids and chip experiments in passages 2–6.

### 2.5. Establishing Microvascular Network

The establishment of a microvascular network was performed according to Sarkanen et al. 2012 with few modifications [21]. The hASCs were seeded in 24-well plates (Nunc, Thermo Fisher Scientific) at a density of 20,000 cells/cm^2^. After 1–2 h, HUVECs at a density of 4000 cells/cm^2^ were seeded on top of attached hASCs. A microvascular network was formed through self-assembly of HUVECs and hASCs without additional biomaterial. Cells were seeded and co-cultured in angiogenic EGM-2 medium where fetal bovine serum was replaced with 2% HS (Biowest). Microvascular network was formed until day 4–7 prior to addition of hepatocytes (details in Appendix A).

### 2.6. iPSC-HLC Spheroid Formation and Culturing

The spheroids were created similarly as described by Kiamehr and Verfaillie [22] with small modifications. At the hepatic differentiation stage, around day 11, when the cells reached the hepatoblasts stage, they were detached by TrypLE (Thermo Fisher Scientific), collected in medium, centrifuged 150× *g*, 5 min, washed with 10% sucrose, centrifuged again 150× *g,* and resuspended in 10% sucrose for a final cell suspension volume of 100 μL. This was mixed with 100 μL of Puramatrix suspension [70 μL Puramatrix (Corning, Corning, NY, USA), 10 μL 10% sucrose, and 20 μL of cold Collagen I (Gibco)]. Spheroids were then created by pipetting 5 μL of this mixture into the hepatocyte maturation medium (3 spheroids/48-well plate well). The medium was changed a few minutes after making the spheroids and again after one hour; then, all medium was changed every other day. The experimental outline for 2D and 3D cultures is presented in Figure 1.

### 2.7. Culturing Cells on a Microfluidic Chip

We first performed preliminary tests with HepG2 cells (ATCC-HB-8065, ATCC, Manassas, VA, USA) to find a suitable cell concentration and biomaterial for culturing cells on microfluidic chips (DAX-1, AIM Biotech, Singapore, Singapore). We tested Geltrex, fibrin and collagen I in various concentrations and cell-to-biomaterial ratios. Experimental details can be found in the Appendix A. In short, we tested Geltrex (Gibco) in two Geltrex-to-cell ratios 3:2 and 2:1, and three different HepG2 cell concentrations: 8, 12, and 20 million cells/mL. Fibrin was tested with three different fibrinogen concentrations: 20, 10, and 5.0 mg/mL. Finally, HepG2 cells were cultured with three different collagen I concentrations: 2.0, 1.0, and 0.5 mg/mL. As Puramatrix (Corning) worked well for the iPSC-HLC spheroid experiments, we also tested that for on-chip culturing, but instead of HepG2 cells, we used HUVECs that were available at the time. The cell suspension (10 million cells/mL) was mixed with Puramatrix in a 1:1 ratio, injected into two microfluidic chips, and cultured in EGM-2 (Lonza) medium for 6 days. To enhance the diffusion of fresh medium to the cell culture area, medium was replaced daily by introducing differing volumes, 90 µL and 50 µL, to the medium reservoirs of each medium channel. After the preliminary biomaterial tests with HepG2 and HUVECs, we performed subsequent on-chip culturing tests with iPSC-HLCs. The differentiating iPSC-HLCs were plated on the chips at the end of stage 2 of the differentiation (Figure 1). Details of the experiments can be found in Table 1 and Appendix A. Samples for RNA, ICC staining, and ELISA analyses were collected at time points d8 and d20. We used the same biomaterials as in the preliminary tests with HepG2 cells, i.e., Geltrex, fibrin, and collagen I, as well as a mix of fibrin–collagen I (Table 1).

### 2.8. Culturing the 3D iPSC-HLC Spheroids in Inserts with Vasculature

The iPSC-HLC spheroids were created as described in the Section 2.6 and then placed in cell culture inserts with vasculature on the bottom of each well of a 24-well plate. Control spheroids were also cultured in inserts but without vasculature. Medium was changed every other day and spheroid samples were collected at spheroid days (differentiation day): d8 (d19), d14 (d25), and d20 (d31). Altogether, three separate experiments (SV1-3) were performed with slightly different settings, the details of which can be found in Appendix A.

### 2.9. iPSC-HLC + HUVEC Spheroids

We combined HUVECs (p6) with iPSC-HLCs in spheroids in a 1:4 ratio to see whether adding ECs into the system would enhance the liver functionality, i.e., increase the expression of liver-specific genes and proteins. These combination spheroids were cultured in medium containing 50% HCM (Lonza) supplemented with 25 ng/mL of HGF (Invitrogen, Thermo Fisher Scientific) and 20 ng/mL OSM (R&D systems) and 50% EGM-2 (Lonza). Samples for RNA were collected at spheroid days 8, 14, and 20, corresponding to differentiation days 19, 25, and 31, respectively.

### 2.10. iPSC-HLCs with HUVECs on Microfluidic Chips

Along with the iPSC-HLC monoculture biomaterial tests on microfluidic chips, we explored co-culturing by adding HUVECs into the chips. We had two different approaches: (a) plating the iPSC-HLCs in Geltrex into the chip and then coating the flanking media channels with HUVECs (exp 7 and 8), or (b) mixing the GFP-HUVECs (p3) with the iPSC-HLCs in Geltrex and plating them into the chip gel channel together (exp 9) (Table 1). Details of the HUVEC coating protocol can be found in the Appendix A. The co-culture chips were kept in mixed culture medium, a 1:1 mixture of HCM and EGM-2 (Lonza), supplemented with 25 ng/mL HGF and 20 ng/mL OSM.

### 2.11. Multilineage Cell Cultures

#### 2.11.1. Medium Tests for Multilineage Spheroids

To create multilineage spheroids, iPSC-HLCs were mixed with HUVECs and hASCs in the ratio of 20:5:1. The isolation and culturing of HUVECs and hASCs are described in Section 2.3 and Section 2.4. First, we performed a medium test experiment with the multilineage spheroids to find the optimal medium composition for further experiments.

We tested three different medium combinations: (1) 100% HCM, (2) 50:50 HCM:EGM-2 (w/o EGF), and (3) 75:25 HCM:EGM-2 (w/o EGF). We collected RNA samples at spheroid (differentiation) days d0 (d11), d7 (d18), and d12 (d23), and performed qPCR to study the expression of *AFP*, *ALB*, *APOA1*, *APOB*, *ASGR1*, *CYP3A5*, *MRP2*, and *CD31*. *GAPDH* was used as the endogenous control. In addition to the multilineage spheroids, we also had iPSC-HLC monoculture spheroids in this experiment as a reference.

#### 2.11.2. Further Multilineage Spheroid Experiments

Subsequent multilineage spheroid experiments were set up to determine whether mixing the three cell types would enhance the liver functionality of the spheroids compared to the iPSC-HLC monoculture spheroids. We used the same cell ratios in all three experiments, i.e., 20:5:1 of iPSC-HLC:HUVEC:hASC. The multilineage spheroids were cultured up to 26 days in 100% HCM alongside the iPSC-HLC monoculture spheroids (controls).

#### 2.11.3. Multilineage Chip Cultures

Lastly, we created multilineage chips by culturing a mix of iPSC-HLCs, HUVECs, and hASCs in fibrin (fibrinogen 10mg/mL, thrombin 2 IU/mL) on-chip using similar cell ratios as with the multilineage spheroids, i.e., 20:5:1. The medium used in these experiments was HCM with growth factors (25 ng/mL HGF and 20 ng/m OSM) and 5 μM aprotinin. Samples were collected at d8, d14, and d20 of on-chip culturing.

### 2.12. Assessing Biomaterial Integrity on the Chip Cultures

The integrity of the biomaterial on the chip cultures was examined by adding 2.0 µm fluorescent microspheres (Invitrogen, Thermo Fisher Scientific) into one medium channel and visually observing whether they crossed the gel channel to the opposite medium channel. The microsphere stock solution was first diluted 1:1000 with sterile water, and then further 1:1 with warm medium. Next, 90 µL of the microsphere mixture was added to the top port and 50 µL to the bottom port to create a hydrostatic pressure and an interstitial flow across the hydrogel area. The gels and medium channels were immediately observed and imaged with EVOS FL imaging system (Thermo Fisher Scientific). Afterwards, the mixture was replaced with regular medium and cell culturing was continued.

### 2.13. Assessing Cell Viability

#### 2.13.1. Cell Viability in the Spheroid Cultures

Cell viability during the spheroid cultures was assessed with the ReadyProbes™ Cell Viability Imaging Kit (Invitrogen, Thermo Fisher Scientific). A drop of each stain, one for cell nuclei and one for dead cells, was added to 500 µL of media with the samples and incubated at +37 °C for 20 min. Samples were then imaged with the EVOS FL imaging system (Thermo Fisher Scientific).

We also used Calcein-AM, which stains live cells green, and ethidium homodimer-1 (EthD), which stains dead cells red by entering cells with plasma membrane damage. Samples were washed twice with PBS and a working solution of 1:8000 of Calcein-AM and EthD was added. Samples were incubated at 37 °C for 30 min in the dark and imaged with a EVOS FL microscope (Thermo Fisher Scientific).

#### 2.13.2. Cell Viability in the Microfluidic Chip Cultures

The ReadyProbes™ Cell Viability Imaging Kit (Invitrogen, Thermo Fisher Scientific) was also used to assess cell viability during the chip culture experiments. Two drops of each stain were mixed with 1 mL of medium and the mixture was pipetted into the chips with differing volumes on each side to create a flow through the gel. The chips were incubated 10 min at +37 °C followed by 10 min at RT. The images were taken with an EVOS FL microscope (Thermo Fisher Scientific).

### 2.14. RNA Extraction and Real-Time Quantitative PCR (qPCR)

#### 2.14.1. Spheroids

Spheroids were lysed in 500 μL Qiazol lysis buffer and stored at −80 °C until RNA extraction, which was conducted using the miRNeasy mini kit (2D samples) or micro kit (3D samples) according to the manufacturer’s recommendations and including the DNAse treatment (Qiagen, Hilden, Germany). Three spheroids were used for each of the four biological replicates. RNA was eluted in 40 μL (2D) or 14 μL (3D) of sterile water.cDNA was prepared from 6 μL of RNA using the High-Capacity cDNA Reverse Transcription Kit (Thermo Fisher Scientific). TaqMan gene expression assays (Thermo Fisher Scientific) were used in quantitative real-time PCR, which was performed using a Quant Studio 12K Flex Real-Time PCR system (Thermo Fisher Scientific). The expression of each gene was normalized to the expression of the endogenous control GAPDH, and fold change was calculated with the ΔΔCt method [23] relative to spheroid day 8 (diff. day 19).

#### 2.14.2. Microfluidic Chips

Cells from the chip cultures were collected by lysing the cells and biomaterials in 700 μL of Qiazol on the chip. In the last multicellular chip experiment, to enhance the sample lysis, a 5 min pretreatment with 135 μL of 10 mg/mL pronase (Merck Millipore) was performed before lysing everything in 540 μL of Qiazol. RNA extraction (Qiagen miRNeasy micro kit), cDNA preparation, and qPCR were performed similarly as for spheroid samples (see Section 2.14.1). RNA from chip experiments was collected at time points d8, d14, and d20. Chip day 8 (diff. day 19) samples were used as reference.

### 2.15. Staining

#### 2.15.1. Immunocytochemical (ICC) Analysis of 2D iPSC-HLC Cultures

Protein expression of liver-specific markers in 2D control cultures was determined with immunocytochemical staining, according to the protocol presented in Appendix A. The antibodies are listed in Appendix A.

#### 2.15.2. Immunohistochemical (IHC) Analysis of Spheroid Sections

The spheroids were first fixed in 4% PFA and mounted in paraffin before cutting 4 μm sections with a sectioning device (Leica SM2010 R microtome; Leica Biosystems, Wetzlar, Germany). The sections were dewaxed before staining with antibodies against various liver, cholangiocyte, and EC markers (Appendix A). The details of the protocol can be found in the Appendix A. The stained spheroid sections were imaged with an Olympus IX51 Fluorescence microscope (Olympus Corporation, Hamburg, Germany) and ImageJ was used for further image analysis.

#### 2.15.3. Hematoxylin-Eosin Staining of Spheroid Sections

The spheroid sections were first incubated 30 min at 60 °C, then stained using a KEDEE KD-RS3 Slide Stainer (Kedee Instruments, Zhejiang, China). After staining, the sections were incubated in xylene 2 × 5 min at RT, mounted with a Dako coverslipper (Agilent, Santa Clara, CA, USA), and left to dry overnight. Imaging was conducted using the Zeiss Axio Scope A1 Fluorescence microscope (ZEISS, Jena, Germany).

#### 2.15.4. Immunocytochemical Analysis of Cells Cultured on Microfluidic Chips

Cells on the chip cultures were fixed in 4% PFA and stained with the same antibodies as the spheroids (Appendix A). Details of the chip staining protocol are presented in the Appendix A. The stained chips were imaged with an Olympus IX51 Fluorescence microscope (Olympus Corporation), and ImageJ was used for further image analysis.

### 2.16. Media Analyses

#### 2.16.1. Albumin

The Human Albumin ELISA Kit (Abcam, Cambridge, UK) was used to measure the albumin production by the spheroids from 48 h conditioned medium as a function of time, i.e., the measurements were performed for the same cell culture well at each time point to see the change in ALB levels from one time point to the next. In the spheroid experiment with vasculature, ALB levels were measured from two independent experiments, each containing three replicate wells.

From the chips, the medium samples were collected 24 h after the last medium change by pooling all three chip sites, i.e., six medium channels, from one chip into one at time points d8, (d14), and d20.

#### 2.16.2. Urea

The QuantiChrom Urea Assay Kit (BioAssaySystems, Hayward, CA, USA) was used to measure 48 h urea production in 3D spheroids and 2D culture systems. The values from the 3D spheroid culture wells were multiplied by three to make them comparable to the 2D cultures regarding the cell number in both systems. In the insert culture experiments (Section 2.8.), we only compared the 3D spheroids with and without the vasculature, so the values were not multiplied.

### 2.17. Statistical Methods

Statistical analysis was performed with GraphPad Prism 9 statistical software (GraphPad, San Diego, CA, USA). The results are presented as mean + SEM or SD from at least three experiments when applicable. Statistical analyses comparing gene expression levels in more than two groups were performed with 2w ANOVA and *p* < 0.05 was considered statistically significant. Multiple comparisons were performed by Šídák’s or Tukey’s tests as suitable. Statistical analyses comparing the immunofluorescence staining intensities between time points were performed with 1w ANOVA and *p* < 0.05 was considered statistically significant. Multiple comparisons were performed by Tukey’s tests.

### 2.18. The Selective Plane Illumination Microscopy (SPIM) Imaging

We used SPIM to obtain a detailed 3D view of the 3D iPSC-HLC and HLC+HUVEC spheroids at spheroid d20, and multilineage spheroids at d14. SPIM technology is a fast, minimally invasive optical sectioning method for 3D imaging of fluorescing samples. A thin laser light sheet is focused into the specimen, taking two-dimensional images of the illuminated slice with a perpendicularly positioned detector (sCMOS camera). The 3D stacks were obtained by moving the spheroid specimen orthogonally across the light sheet between consecutive images and the sample was rotated to collect 3D stacks from multiple angles (views) [24].

### 2.19. The Optical Projection Tomography (OPT) and Electrical Impedance Tomography (EIT) Imaging

The spheroids were imaged with a combined OPT-EIT technique [25] at room temperature at time points d0 (iPSC-HLC, parallel samples (n) = 3; iPSC-HLC+HUVECs, n = 3), d14 (iPSC-HLC, n = 2; iPSC-HLC+HUVECs, n = 2), and d20 (iPSC-HLC, n = 1; iPSC-HLC+HUVECs, n = 3). The OPT-EIT technique simultaneously acquired both optical and electrical data that were used to reconstruct 3D images. OPT was used in brightfield mode to obtain spheroid morphologies and locations. OPT reconstructions were further segmented, and the morphological information was fused into EIT reconstructions. This data fusion enabled us to accurately reconstruct the electrical conductivity of the spheroids.

## 3. Results

### 3.1. The 3D iPSC-HLC Monoculture Spheroid and Microfluidic Chip Cultures

#### 3.1.1. Mature Liver Marker Gene and Protein Expression Was Increased in the 3D iPSC-HLC Monoculture Spheroids during Culture

During the 2D culture, the expression of most studied genes was significantly down-regulated from d19 onwards (Figure 2A). This was also seen at the protein level when staining for AFP, ALB, A1AT, and CK19 (Appendix A). This suggests that the best iPSC-HLC maturity was reached around d19 in the 2D cultures.

In the 3D iPSC-HLC spheroid cultures, the expression of all studied genes rose towards spheroid d20 (diff. d31) and persisted longer than in 2D cultures (Figure 2B). Interestingly, *ALB* gene expression was statistically significantly up-regulated at spheroid d20 (diff. d31; Figure 2B), which was also seen at the protein level in the immunohistochemical staining of spheroid sections (Figure 3A). Moreover, CK19 protein was detected at all studied spheroid time points, and A1AT increased until spheroid d20. MRP2 staining was observed at all time points. The SPIM imaging showed strong expression of AFP, ALB, and CK19, as well as the cuboidal cell phenotype of the iPSC-HLCs at spheroid d20 (Figure 3B). The H&E staining at spheroid d26 (diff. d37) showed ductal-like structures (Figure 3C).

#### 3.1.2. Albumin Production was Increased in the 3D Spheroids over Time Whereas Urea Stayed Constant

In agreement with *ALB* gene expression and immunostaining results, ALB production into cell culture medium decreased in the 2D cultures of iPSC-HLCs over time. On the contrary, in the 3D iPSC-HLCs spheroid cultures, the ALB production increased (Figure 4A). Due to large variation, however, statistical significance was not detected. Urea production stayed constant through the experiments both in 2D and 3D but was higher in 3D spheroids (*p* < 0.0001; Figure 4B).

#### 3.1.3. Biomaterial Testing for 3D on-Chip Hepatocyte Monocultures

We first performed biomaterial testing for on-chip culturing with HepG2 and HUVECs, the detailed results of which can be found in the Appendix A. Then we moved on to iPSC-HLC on-chip culture tests (Table 1). The lower Geltrex-to-cell ratio (3:2) was, unlike with HepG2 cells, not enough to support the 3D culture, and thus the higher Geltrex-to-cell suspension ratio (2:1) was used next. The cells mostly remained in 3D and resembled the preliminary tests with HepG2 cells. However, the gel was hard to handle, and a bead test (at d15/d16 of on-chip culture) revealed gel disintegration at the ends of the gel channels in both Geltrex experiments.

The first fibrin experiment with iPSC-HLCs was terminated prematurely at d8 due to progressive material disintegration. With the addition of 5 µM aprotinin to the culture medium, the material stayed stable throughout the experiment, and even though the cells formed some elongated colonies inside the gel, the cells remained in 3D. The collagen I chips were stable at d0, yet showed signs of material decay at d7, and also prominent cell death throughout the experiment. Thus, also collagen I was excluded. Lastly, we tested fibrin–collagen I with the addition of 5µM aprotinin to the culture medium. The material was highly viscous and thus difficult to handle, and the adjustment of pH to an appropriate range proved challenging. Notably, the gel started to shrink and show signs of degradation around d14 of on-chip culturing.

The qPCR data suggest that at d8 of on-chip culture, Geltrex chips had the highest expression of all studied genes except *AFP*. Expression levels of all studied genes, except *MRP2*, dropped at d20 of on-chip culture; then, the chips with Geltrex had the highest expression of *ALB* and *CYP3A5* (Appendix A).

### 3.2. The 3D iPSC-HLC Spheroids in Insert Culture with Vasculature

We wanted to test whether culturing the iPSC-HLC spheroids in cell culture inserts together with vascular structures formed by HUVECs and hASCs in the bottom of the wells would enhance the liver-specific phenotypes of these spheroids.

#### 3.2.1. Gene Expression

In the spheroids cultured in inserts with vasculature, there was a trend of decreasing expression from d8 to d20 for most studied genes, and statistical significance was shown for *APOA1* and *APOB* (Appendix A), similarly as seen before in the 2D cultures (Figure 2A).

Liver-specific gene expression in the iPSC-HLC control spheroids in the insert experiments (Appendix A) only partly followed the pattern of the previous 3D iPSC-HLC spheroid experiments (Figure 2B). *ALB* expression stayed constantly (Appendix A) opposite to the increase detected in the previous 3D spheroid experiment (Figure 2B). This is probably due to the mixed medium (EGM:HCM), which only partially consists of hepatocyte maturation medium. The effect on albumin was also detected when testing different medium compositions for iPSC-HLC monoculture and multilineage spheroids (Appendix A). There was a statistically significant decrease in *APOB* from spheroid d8 onwards. Expression of *MRP2* increased first from d8 to d14, but then decreased statistically significantly from d14 to d20 in the control spheroids (Appendix A).

#### 3.2.2. Albumin and Urea Production

A somewhat higher level of ALB production was detected in the spheroids that were cultured in inserts with vasculature compared with the spheroids cultured alone (cntrl), but the difference was statistically significant only in the spheroid at d8 (overall *p* < 0.01; Appendix A). Urea production appeared slightly higher for spheroids cultured with vasculature, but the difference was not statistically significant (Appendix A).

### 3.3. The 3D Co-Culture of iPSC-HLCs and HUVECs

Next, we explored whether adding one supportive cell type, i.e., endothelial cells (HUVEC) into the cultures together with iPSC-HLCs would promote the formation of primitive vascular structures and potentially enhance the expression of liver-specific genes and proteins in the cultures. First, co-culture iPSC-HLC and HUVEC spheroids were created. Then, HUVECs were cultured alongside iPSC-HLCs on microfluidic chips. Moreover, we utilized the OPT and impedance measurements to further characterize the iPSC-HLC+HUVEC spheroids with a novel 3D technique.

#### 3.3.1. Gene Expression in Co-Culture Spheroids

Expression of *AFP* peaked at spheroid d14 (diff. d25) for both spheroid types, but the increase from d8 to d14 was statistically significant only in the iPSC-HLC+HUVEC spheroids. The *ALB* gene expression increased in both spheroid types from spheroid d8 (diff. d19) onwards, and in the iPSC-HLC+HUVEC spheroids even until spheroid d20 (diff. d31; Figure 5A,B). *APOB* and *ASGR1* decreased statistically significantly from spheroid d8 to d20 in the iPSC-HLC spheroids, but not in the co-culture spheroids. *MRP2* and *CYP3A5* did not change statistically significantly in either spheroid type (Appendix A). 

#### 3.3.2. The OPT, EIT, and SPIM Imaging and Immunocytochemical Staining Reveal Decreasing Spheroid Size, Increasing Conductivity, and Persisting AFP Expression

The OPT showed that both the iPSC-HLC monoculture and iPSC-HLC+HUVEC co-culture spheroid volume decreased (from 2 μL to 0.4 μL) from spheroid d0 to d20 (diff. d11 and d31, respectively; Appendix A). The OPT-EIT results suggest that the conductivity increased from spheroid d0 to d14 for both spheroid types (Appendix A), but the increase was statistically significant only for the iPSC-HLC spheroids (d0, 24.6 ± 12.5 mS/cm vs. d14, 130.7 ± 50.5 mS/cm, *p* < 0.05). A snapshot of an OPT video of two iPSC-HLC+HUVEC co-culture spheroids at d0 is presented in Figure 5C. The d20 spheroids were so small (volume fraction in imaging chamber was <0.3%) that the EIT measurements were not considered reliable, and thus the results are not shown. Conductivity of the cell culture medium was 15 mS/cm (conductivity meter, Hanna Instruments, HI-8733), thus possible medium diffusion did not induce the detected increase in conductivity. The increase in conductivity from d0 to d14 could indicate change in cell number, their organization, or possibly cell death.

Immunocytochemical staining of the iPSC-HLC+HUVEC spheroid sections indicated strong expression of ALB and hepatocyte progenitor/cholangiocyte marker SOX9 as well as large empty areas devoid of cells or biomaterial at spheroid d14 (Figure 5D), which could partly explain the increasing conductivity values at d14 detected in the OPT-EIT measurements. The endothelial cell marker CD31 staining was strong at all time points in the iPSC-HLC+HUVEC spheroids (Appendix A). AFP and MRP2 were expressed to some degree in both iPSC-HLC and iPSC-HLC+HUVEC spheroids at all time points (Appendix A), mirroring the gene expression levels.

The SPIM imaging was conducted at spheroid d20 (diff. d31) and showed expression of ALB and MRP2 in the cells within the spheroid (Figure 5E). There was also strong expression of AFP throughout the spheroid (Figure 5E), similar to that earlier detected in the iPSC-HLC monoculture spheroids (Figure 3B). The AFP antibody seemed to stain the biomaterial to some degree, but there was also persisting AFP expression in the iPSC-HLCs, which suggests that the iPSC-HLCs possess a somewhat immature phenotype. The CD31 expression was rather sporadic inside the spheroid (Figure 5E), and no vessel-like structures were seen.

#### 3.3.3. iPSC-HLCs in 3D Co-Culture with HUVECs on Microfluidic Chips

Along with the biomaterial tests with iPSC-HLC, we performed on-chip experiments where we added HUVECs with iPSC-HLCs in the chip gel channels. The number of analyzed chips was too low to reach statistical significance, so we can only see trends. At d8 of on-chip culture, *AFP* and *ASGR1* gene expression appeared higher in the co-culture chips. The expression of the other studied genes was higher in the iPSC-HLCs monoculture chips at d8. However, at d20, expression of all studied liver-related genes was higher in the co-culture chips compared to iPSC-HLCs monoculture chips (Appendix A). The ALB ELISA results are presented in Section 3.4.4.

Immunocytochemical stainings of the chips from exp 2, 8, and 9 (Table 1), i.e., iPSC-HLC monocultures (exp 2), iPSC-HLCs in the gel channel and HUVECs lining the media channels (exp 8), and iPSC-HLCs and HUVECs co-cultured together in the gel channel (exp 9) can be seen in Figure 6. When HUVECs were cultured in the media channels flanking the gel channel with iPSC-HLCs, the latter seemed to form tubular structures by d20 on-chip, more than in iPSC-HLC monocultures or when the HUVECs were mixed with iPSC-HLCs in the gel channel. In the last experiment, the iPSC-HLCs seem to be migrated from the gel channel towards the flanking media channels (Figure 6).

### 3.4. Multilineage Cultures of iPSC-HLCs, HUVECs and hASCs

In the preliminary medium tests, 100% HCM worked well regarding the liver-specific gene expression (Appendix A). Admittedly, *CD31* (endothelial cell marker) decreased from d7 to d12 in the multilineage spheroids, but that happened equally in all tested medium combinations (Appendix A). The 100% HCM medium seemed to endorse the highest expression of *ALB* in the multilineage spheroid at spheroid d12 (Appendix A), and thus it was chosen for all subsequent multilineage culture experiments. Detailed information about the preliminary medium tests and their results can be found in the Appendix A. For multilineage chip experiments, 5 mM aprotinin was added to the medium to minimize the amount of biomaterial decay during the culture period. 

#### 3.4.1. Gene Expression in Multilineage Spheroids

In the multilineage spheroids, the expression of most studied genes was significantly down-regulated from spheroid d8 (diff. d19) onwards, but not *ALB, MRP2,* or *ANGPT2* (a marker for early stage vascular formation; Figure 7A). Decreasing *AFP* and continuous expression of *ALB* suggests hepatocyte maturation, but decreased expression of other liver-enriched genes *APOA1* and *APOB* (apolipoproteins related to lipid metabolism), *ASGR1* (liver-specific cell surface marker), and *CYP3A5* (drug metabolism) suggests declining functionality in the multilineage spheroid culture. EC marker *CD31* was statistically significantly down-regulated in the multilineage spheroids.

In the iPSC-HLC monoculture spheroids, the expression of *ALB* increased statistically significantly until spheroid d20 (diff. d31), other changes did not reach statistical significance, but there is also a trend of increasing expression for, e.g., *CYP3A5* and *ASGR1* (Figure 7B), altogether suggesting that the iPSC-HLC mature and stay functional in the 3D monoculture spheroids.

#### 3.4.2. Protein Expression in Multilineage Spheroids

The IHC staining of multilineage spheroids showed similar results as the gene expression studies: AFP decreased, ALB and MRP2 appeared constant, and CD31 was down-regulated towards later time points (Figure 8). The SPIM imaging showed sporadic staining of AFP and CD31 at spheroid d14 (Appendix A), similarly as the IHC staining (Figure 8). Additionally, we detected strong expression of liver-specific A1AT at all time points, and hepatocyte progenitor/cholangiocyte marker CK19 more pronounced at the earlier time points d8 and d14 whereas SOX9 is visible at spheroid d20 and d26 (Figure 8).

#### 3.4.3. Multilineage Chip Cultures

In the multilineage chip experiments, we used the same cell ratios on the microfluidic chips, i.e., iPSC-HLC: HUVEC: hASC in 20:5:1, as we did in the multilineage spheroids. The qPCR results show that expression of *AFP* decreased significantly during on-chip culture (Figure 9). The *ALB* expression increased from d8 to d14, and by d20 on-chip, the expression returned to d8 level. We saw a statistically significant decrease in the expression of *CD31* (endothelial cell marker) from d14 to d20. *ASGR1* (hepatocyte surface marker) or *ANGPT2* (early marker for angiogenesis) did not change statistically significantly. There was a trend of decreasing expression of *APOA1*, *APOB,* and *CYP3A5* from d8 onwards (Figure 9).

The immunocytochemical staining results of chips at d8, d14, and d20 of on-chip culture mirrored the gene expression results. The AFP protein expression was clearly down-regulated, whereas ALB stayed constant (Figure 10). A1AT and MRP2 were also constant at all studied time points. Interestingly, the MRP2 antibody seemed to stain the hASCs in the culture. The endothelial cell marker CD31 protein expression appeared strongest at d14 of on-chip culture, as did the hepatocyte progenitor/cholangiocyte marker CK19, which was then down-regulated towards d20 of on-chip culture.

#### 3.4.4. Albumin Production Stayed Constant in the Multilineage Chip Cultures

The albumin production results of all microfluidic chip cultures showed that the secretion of albumin decreased from d8 to d20 of on-chip culture for the iPSC-HLC monoculture chips in Geltrex and fibrin as well as in iPSC-HLC chips with HUVECs mixed with iPSC-HLCs in the gel channel (Figure 11). In the fibrin–collagen chips, ALB production was very low at both time points (similarly as the *ALB* gene expression levels (Appendix A)). However, in the multilineage chips, the albumin production stayed constant from d8 to d20 of on-chip culture (Figure 11), mirroring the gene expression levels (Figure 9).

## 4. Discussion

One of the biggest limitations of the current in vitro liver systems is the immature phenotype of the induced pluripotent stem cell-derived hepatocyte-like cells (iPSC-HLCs) compared with primary human hepatocytes (PHHs) or liver tissue [26]. Hence, improved protocols are needed to enhance the maturity and functionality of the iPSC-HLCs, increasing their usability in in vitro models. Moreover, in 2D cultures, the microenvironment lacks physical cues from the extracellular matrix (ECM) on cell behaviour. To address this limitation, various cell culture techniques and models were developed. One such approach is the use of ‘sandwich cultures’, where cells are cultured between layers of ECM substrates, more closely mimicking the native tissue architecture, and promoting enhanced cell functionality. Other 3D culture systems, such as scaffold-based or spheroid cultures, also aim to promote cell–cell interactions and functionality to closely mimic that of in vivo conditions [2].

In our study, we utilized iPSC-HLCs to create in vitro liver models and studied the effects of 3D culturing and addition of non-parenchymal cells (NPCs) on the maturation of the iPSC-HLCs. Importantly, all cells used in this study were of human origin, which is relevant when using the model for, e.g., drug metabolism or toxicity studies, since pathways related to these differ between rodents and humans [27]. Thus, our 3D in vitro models hold the potential to serve as valuable tools in the development of more complex and human relevant systems to assess, e.g., hepatotoxicity in drug discovery.

For our iPSC-HLCs, we used a three-step differentiation protocol of which the first two stages were performed in 2D and then the immature iPSC-HLCs were transferred with hydrogel to the chips or moulded into spheroids after stage 2 of the differentiation protocol, i.e., at d11 of differentiation (Figure 1). We used hydrogels with the cells to create the 3D model either as spheroids or on microfluidic chips for stage 3 of differentiation rather than relying on self-aggregation of cells or just inserting them on chips as such. We successfully extended the 3D culturing of iPSC-HLCs up to d20 on-chip or even d26 as spheroids corresponding to differentiation d31 and d37, respectively.

Hydrogels, such as Puramatrix, mimic ECM properties and are thus ideal as scaffolds for in vitro 3D liver models [28]. There are only a few studies on Puramatrix and hepatocytes; one used Puramatrix as a reference material for HepG2 and HepaRG self-aggregation studies [29] and another for Hep3B encapsulation on a micropillar [30]. Previous spheroid studies mostly relied, e.g., on self-aggregation of iPSC-HLCs in ultralow attachment plates [31] or microwell cell culture devices [4] while we used iPSC-HLCs and Puramatrix to create spheroids of uniform size by pipetting as described by Kiamehr and Verfaillie [22]. In our study, 3D spheroid culturing outperformed the traditional 2D culture system, as expected. This is most probably mainly due to dedifferentiation and cell death in the 2D cultures during extended culturing. Our qPCR results clearly show that extending the traditional culture time of 20 days to 25, 31, and even 37 days, resulted in decreased expression of liver-specific genes. During the 3D spheroid culturing, on the other hand, their expression was increased until spheroid d20, which equals to differentiation d31. The gene expression results are further supported by ELISA analyses of albumin and urea production. Moreover, the staining of the 2D cultures showed cell deterioration, which often happens during extended 2D cultures. Overall, our results agree with previous studies demonstrating improved hepatic maturation of iPSC-HLCs in 3D aggregates [4,31,32,33].

Next, we utilized an insert culture system to study whether culturing spheroids with vasculature induced by co-culturing HUVECs with hASCs would promote liver functionality when compared to iPSC-HLC spheroids cultured (in inserts) alone. We did not see improved liver gene expression, but did see a trend of higher ALB and urea production by the spheroids cultured in inserts with vasculature compared to spheroids cultured alone. This differs somewhat from the results of a previous study, which show paracrine signals to induce hepatocyte differentiation [34]. In that study, however, the hepatic-specified endoderm iPSCs formed a 2D monolayer, which then differentiated to hepatocytes when cultured in separate chambers with HUVECs and MSCs. Additionally, we used adipose-derived stem/stromal cells (hASCs), whereas they used human bone marrow-derived cells, which we and others previously showed to differ in their ability to promote the development of vascular structures [12,35]. Moreover, in our model, the effect of co-culture only affected the maturation phase of the iPSC-HLCs. It might therefore be beneficial to start the co-culture of iPSC-HLC spheroids with vasculature earlier to enhance the effect of the paracrine signals also in our 3D spheroid model.

Since the paracrine signaling from vasculature did not seem to strongly promote the maturation of iPSC-HLC spheroids, we decided to add NPC types into the spheroids with iPSC-HLCs. Endothelial cells (EC) represent a major cell type of interest for co-culture models given their fundamental role in lining the blood vessels that supply nutrients and oxygen to tissues. We added ECs in the form of HUVECs into the 3D iPSC-HLC spheroids at differentiation d11 when the spheroids were formed to test whether this would promote the liver-like functionality of the spheroids. We did not detect a clear increase in the expression of liver-specific genes in the iPSC-HLC+HUVEC spheroids, but we did see *ALB* expression increasing longer (until spheroid d20) than in the iPSC-HLC monoculture spheroids. A previous 2D study did not find a supportive effect of HUVECs on the iPSC to hepatic differentiation [13]. On the contrary, another study showed EC co-culturing with iPSC-HLCs to improve hepatic functions [36]. The setup slightly differed from ours as they differentiated HLCs from iPSCs as embryoid bodies with ECs in a 3:1 ratio, whereas we created spheroids later from hepatoblasts and combined them with HUVECs (4:1) and Puramatrix. We saw SOX9 expression in the iPSC-HLC+HUVEC spheroids, which could imply them expressing a liver progenitor-like or cholangiocyte phenotype [37]. It could also be a sign of hypoxia in the spheroids as suggested by a study with HepaRG cells that showed increasing SOX9 expression in hypoxic conditions [38]. The ratio of iPSC-HLC and HUVECs in our spheroids (20% HUVECs) could be optimized to better support the iPSC-HLC maturation. In a study developing an in vitro liver cancer model of hepatocellular carcinoma cells (Huh7), co-culturing with 2% HUVECs was shown to perform best when concentrations from 0 to 20% were tested [39]. When the addition of ECs with iPSC-HLCs did not significantly improve our model, we decided to create multilineage spheroids by adding a third cell type, i.e., adipose tissue-derived mesenchymal/stromal cells (hASCs). In pioneering studies by Takebe et al. combining HLCs with ECs and mesenchymal stem cells (MSCs), perfusable vasculature within iPSC-derived organoids was induced through transplantation into host animals, where native vasculature penetrated into pre-vascularized organoids [40,41]. We hypothesized that adding ECs and hASCs together could enable formation of vascular structures within the spheroids, and thus endorse the maturation and functionality of the iPSC-HLCs. Contrary to our expectations, however, the multilineage spheroid model did not significantly outperform the iPSC-HLC monoculture spheroids. In the multilineage spheroids, most liver-specific genes were down-regulated from spheroid d8 onwards, but both ALB gene and protein expression did stay high until d26 of spheroid culturing. In the iPSC-HLC monoculture spheroids, the statistically significantly increasing *ALB* expression and persisting expression of other liver-specific genes suggested that the iPSC-HLCs stay functional until spheroid d26, which corresponds to differentiation d37. So the 3D spheroid models, specifically the iPSC-HLC monoculture model, stayed functional considerably longer than traditional 2D models, thus giving them an advantage when performing long-term in vitro studies, but further optimization is needed to increase CYP expression levels, relevant to, e.g., drug clearance studies [42]. The overall underperformance of the multilineage spheroid model in comparison to the monoculture model might be due to the non-optimal growing conditions of the other cell types or too-high cell density leading to, e.g., apoptosis. Additionally, cell–cell interactions in the spheroids might not be optimal for the maturation, as it is generally known that cell–cell and cell–material interactions significantly affect the differentiation (reviewed in [43]). As with the iPSC-HLC + HUVEC spheroids, the co-culture started only at the hepatoblast stage. Ma et al. showed in a bioprinting study that hepatic progenitor cells are better for 3D hydrogel encapsulation and co-culture with ECs and MSCs than more mature HLCs [44]. We propose that starting the co-culture even earlier, after reaching definitive endoderm, might prove advantageous. Furthermore, tissue-specific EC and mesenchymal cells instead of HUVECs and hASCs might better support differentiation of iPSC-HLCs as suggested in a previous 2D in vitro liver study [45]. All in all, based on our results and results of previous studies, it can be concluded that initial cell numbers, the ratio of iPSC-HLCs to ECs and biomaterial, and an early start of co-culturing along with the medium composition affects the level of functionality of in vitro hepatic models.

As highlighted in a recent review [46], non-destructive techniques for monitoring 3D cell cultures need to be developed. Aptly, we tested our in-house-built OPT-EIT imaging technique for monitoring the spheroids during culture. The OPT-EIT imaging revealed that the size of the spheroids decreased from d0 to d20, whereas the conductivity values increased from spheroid d0 to d14. Since Puramatrix is a hydrogel, it is prone to change due to cell activity, which could lead to hydrogel shrinkage as was shown to happen, e.g., with polymeric scaffolds during osteogenic differentiation [47]. The increased conductivities could be due to cell growth and increased cell–cell contacts through gap junctions [48,49]. The gene expression levels peaked at spheroid d14, so we speculate that the increasing conductivity could be due to cell division and/or increased cell–cell contacts. The somewhat lower conductivity values at d14 for the iPSC-HLC+HUVEC than for the iPSC-HLC spheroids could indicate that the cell–cell contacts inside the iPSC-HLC+HUVEC spheroids might be compromised or not as well formed due to a negative effect by the HUVECs on cell–cell connections as suggested previously [39]. It could also be that different gap junctions of iPSC-HLC and HUVECs differ in their ability to form electrical connections due to, e.g., different connexins [50,51], thereby the conductivity is different for iPSC-HLC and iPSC-HLC+HUVEC spheroids. Further studies are needed to clarify the reasons behind increasing conductivity values. Importantly, these preliminary OPT-EIT results of the iPSC-HLC spheroids demonstrate the monitoring functionality of a novel non-invasive 3D imaging technique. In the future, this technique could be utilized in assessing, e.g., tissue growth, the membrane integrity, and cell viability [52,53].

Alongside the spheroid culturing, we studied the possibility of creating a 3D liver model by utilizing microfluidic chips. Our extensive tests with HepG2 cells identified three hydrogels, i.e., Geltrex, fibrin, and collagen I, to support long-term 3D hepatocyte culturing on the microfluidic chip (AIM Biotech). The iPSC-HLC culturing on-chip was best achieved when using fibrin and adding aprotinin in the medium to slow down fibrinolysis [54]. Even though gene expression levels were somewhat higher in Geltrex, due to easy handling and the low cell death of fibrin, it was thus deemed optimal for long-term iPSC-HLC culturing studies. Our experiments clearly showed that HepG2 cells can be cultured in a larger variety of hydrogels than iPSC-HLCs. We showed in our previous study that iPSC-HLCs are metabolically very active [17], which could partly explain the deterioration of biomaterials in our experiments with them. There is a previous study determining optimal fibrin scaffold conditions for HepG2 cells [55], but each cell type requires special optimum conditions of fibrin and, to our knowledge, our study is the first using fibrin to culture iPSC-HLCs in 3D on-chip. We found 10mg/mL fibrinogen, combined with 2IU thrombin (medium supplemented with aprotinin) to work best for iPSC-HLCs on-chip. A very recent study used fibrin to culture PHHs with fibroblasts is cell aggregates [56], thus also supporting its potential as hydrogel for in vitro liver models.

3D culturing was expected to promote cell–cell interactions as our setups enable direct cellular crosstalk and communication, which could result in better recapitulation of in vivo liver functionality. We successfully continued the 3D culturing up to d20 on-chip corresponding to differentiation d31. Our results concur with previous studies, which introduced microfluidic chip culturing for maturation of iPSC-HLC [57,58]. Moreover, cellular cross-talk is important for hepatic differentiation [59], and it was shown that NPCs improve HLC differentiation [45]. MSCs were shown to promote iPSC-hepatoblast maturation [60] through increased protein secretion and via endorsing the formation of tight hepatocyte parenchyma, which highlights the role of cell–cell communication and endogenous growth factor secretion for in vitro hepatocyte maturation [45,60,61]. Hence, we expected adding hASCs into the liver model along with ECs to promote the maturation of iPSC-HLCs as well as extend the longevity of the liver model. Adding just ECs, i.e., HUVECs, in the chips either in the gel channel with the iPSC-HLCs or in the flanking media channels maintained the liver-like phenotype slightly better than iPSC-HLC monoculture chips, specifically when looking at the ALB production. The multilineage chip experiments showed that adding hASCs along the ECs and iPSC-HLCs improves the model regarding decreasing AFP as well as increasing ALB gene and protein expression. The level of ALB production was best maintained in the multilineage chip model compared to the other chips at d20 of on-chip culture. The longevity of our on-chip liver model could be further extended by adding medium flow to continuously provide fresh nutrients and oxygen to the cells.

## 5. Conclusions

In summary, 3D liver models have higher transferability towards in vivo biology and are thus becoming increasingly important in in vitro research. Here, we give insights into hydrogels that can be used in creating 3D in vitro liver models either as spheroids or on microfluidic chips and introduce a non-invasive OPT-EIT technique that can be used to acquire multi-physical 3D images and electrical conductivity of the spheroid model. Our 3D hepatic models, spheroids, and microfluidic chips extend the functional lifetime of the iPSC-HLCs compared to 2D. The iPSC-HLC monoculture spheroid model performed overall best of the spheroid models, whereas in the chip system, the multilineage model with iPSC-HLC, ECs, and hASCs had the best functionality. In conclusion, our 3D in vitro liver models offer promising human-derived platforms that can be used in various liver-related studies and applications, specifically when using patient-specific iPSCs.

## Figures and Tables

**Figure 1 cells-12-02368-f001:**
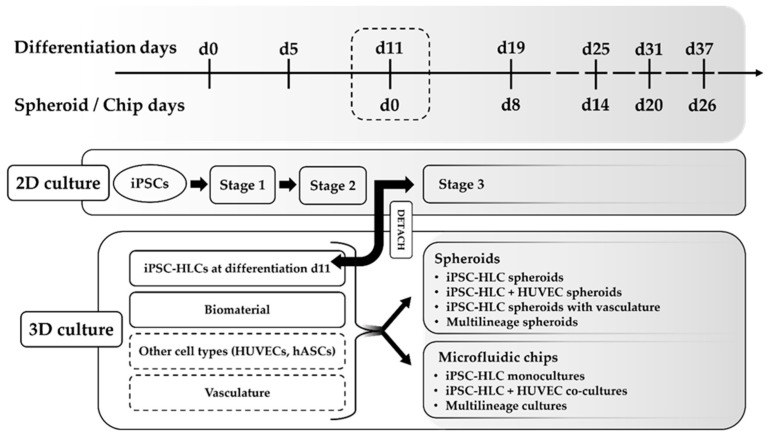
The experimental outline of 2D cultures and 3D spheroid and microfluidic chip cultures. The differentiation protocol of induced pluripotent stem cell-derived hepatocyte-like cells (iPSC-HLCs) consists of three stages. Differentiation d11 corresponds to spheroid/chip d0: at the end of stage 2 (d11), the differentiating cells are detached and combined with a biomaterial to create 3D cultures, as either spheroids or on microfluidic chips. Human umbilical vein endothelial cells (HUVECs), human adipose tissue-derived mesenchymal stem/stromal cell (hASCs), or vasculature were added to some of the 3D cultures together with iPSC-HLCs to create co-cultures or multilineage cultures.

**Figure 2 cells-12-02368-f002:**
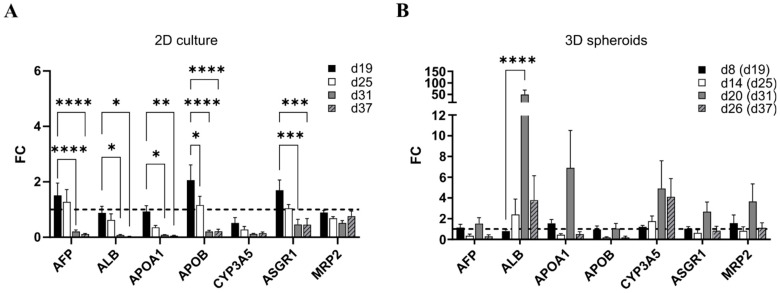
The expression of liver marker genes in 2D culture and 3D iPSC-HLC spheroids. Gene expression in (**A**) conventional 2D culture at differentiation days 19, 25, 31, and 37; and (**B**) 3D spheroids at spheroid days 8, 14, 20, and 26, corresponding to differentiation days 19, 25, 31, and 37, respectively. Fold change (FC) is relative to differentiation day 19 in each culture setting (i.e., spheroid d8 in 3D spheroids), and the dashed line denotes FC = 1. Three separate experiments were performed for both 2D and 3D and from each, and three biological replicates at each time point were collected and run in triplicate in qPCR. Bars represent mean ±SEM. Statistical significance was calculated with two-way ANOVA and *p*-values are from Dunnett’s multiple comparison test, *p*-value * <0.05, ** <0.01, *** <0.001, and **** <0.0001.

**Figure 3 cells-12-02368-f003:**
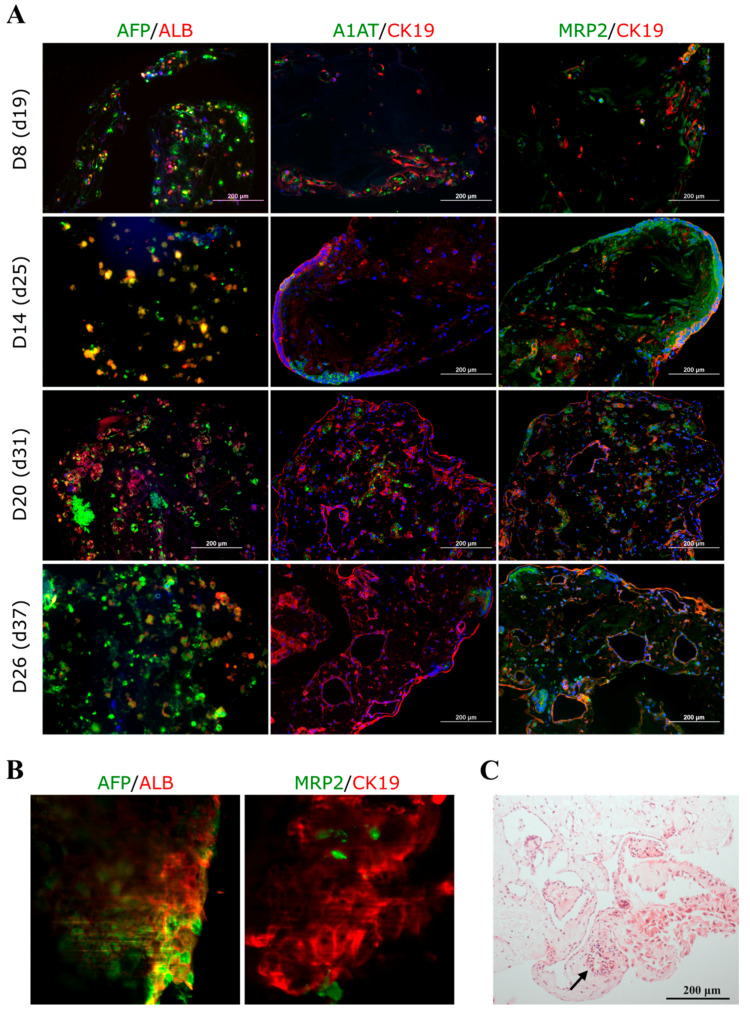
Protein expression in the 3D iPSC-HLC spheroids. (**A**) Spheroid sections were stained at spheroid days 8, 14, 20, and 26, corresponding to hepatocyte differentiation days 19, 25, 31, and 37, respectively. The sections were imaged with an Olympus IX51 Fluorescence microscope (Olympus Corporation), nuclei were counterstained with DAPI (blue). Scale bars 200 µm. Corresponding fluorescence intensity quantification for all markers can be found in Appendix A. The images show hepatocyte markers alpha-fetoprotein (AFP), alpha 1 antitrypsin (A1AT), albumin (ALB), hepatoblast/cholangiocyte marker cytokeratin 19 (CK19), and bile duct marker multidrug resistance-associated protein 2 (MRP2). (**B**) Snapshots of SPIM imaging (zoomed in from the original magnification of 4×) taken at spheroid d20 (differentiation d31). (**C**) A hematoxylin–eosin staining of a spheroid section at spheroid d26 showing ductal-like structures (arrow).

**Figure 4 cells-12-02368-f004:**
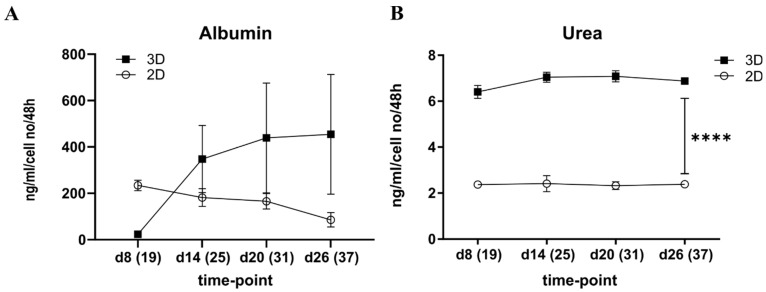
Effect of the cell culture system on albumin and urea production by iPSC-HLCs. (**A**) Albumin production decreased in the 2D culture and increased in the 3D spheroid cultures of iPSC-HLCs from differentiation day 19 to day 37 corresponding to spheroid d8 and d26, respectively. (**B**) Urea secretion stayed constant in both culture systems but was higher in the 3D spheroids compared to the 2D cultures (*p* < 0.0001 at all time points). Some error bars are not shown when the SEM values fall within the symbols. Values are mean ± SEM. **** <0.0001.

**Figure 5 cells-12-02368-f005:**
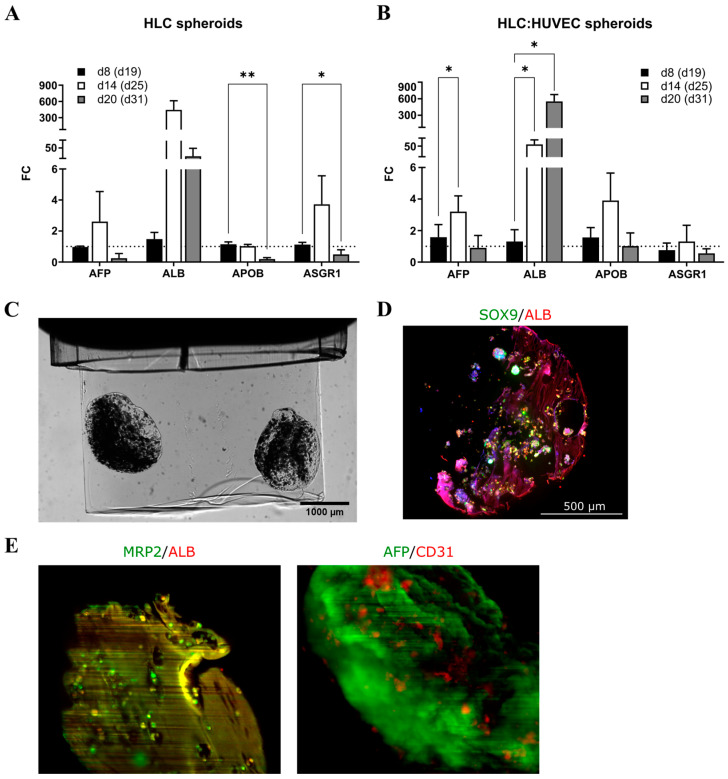
Gene expression, OPT, and SPIM imaging of the iPSC-HLC and iPSC-HLC+HUVEC spheroids. Gene expression levels in the (**A**) iPSC-HLC and (**B**) iPSC-HLC+HUVEC spheroids during culturing at spheroid days 8, 14, and 20, corresponding to differentiation days 19, 25, and 31, respectively. Fold change (FC) relative to spheroid d8 for each spheroid type. Values are mean ± SD. Statistical significance was calculated with two-way ANOVA and *p*-values are from Dunnett’s multiple comparison test, *p*-value * <0.05, ** <0.01. (**C**) Snapshot of an OPT video of iPSC-HLC+HUVEC spheroids at spheroid d0, and (**D**) immunocytochemical staining of an iPSC-HLC+HUVEC spheroid section at spheroid d14. (**E**) Snapshots of SPIM imaging of iPSC-HLC+HUVEC spheroids at d20 (diff. d31). MRP2, multidrug resistance-associated protein 2; ALB, albumin; AFP, alpha-fetoprotein; and CD31, cluster of differentiation 31 (endothelial cell marker).

**Figure 6 cells-12-02368-f006:**
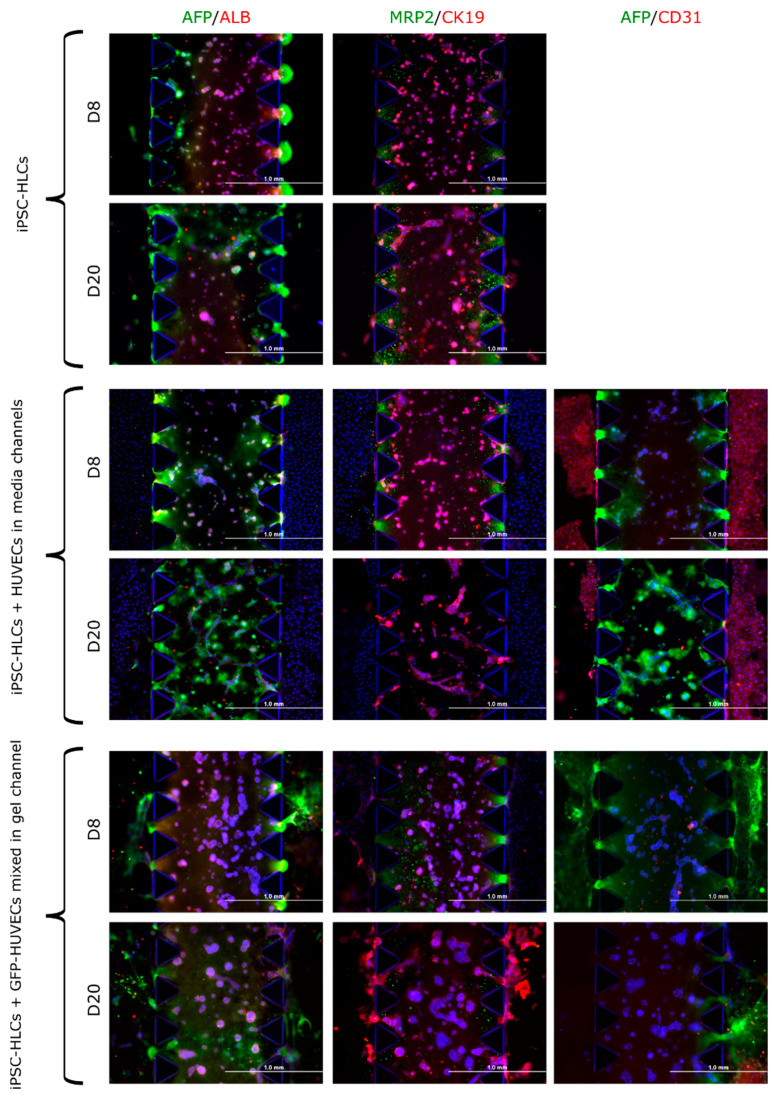
Expression of hepatic and endothelial marker proteins in iPSC-HLCs cultured alone or with endothelial cells (HUVECs) on microfluidic chips (AIM Biotech). First, the iPSC-HLCs were cultured alone (top two rows; exp 2 in Table 1). Next, the HUVECs were plated into the media channels at d4 of iPSC-HLC on-chip culturing (two middle rows; exp 8 in Table 1), and lastly, the GFP-HUVECs were plated together with the iPSC-HLCs in the chip gel channels at d0 (two lowest rows; exp 9 in Table 1). Geltrex was used as the biomaterial in all experiments. The chips were imaged with an Olympus IX51 Fluorescence microscope (Olympus Corporation), nuclei were stained with DAPI (blue). Scale bars 1000 µm. ALB, albumin; AFP, alpha-fetoprotein; MRP2, multidrug resistance-associated protein 2; CK19, cytokeratin 19; and CD31, cluster of differentiation 31 (a marker for endothelial cells).

**Figure 7 cells-12-02368-f007:**
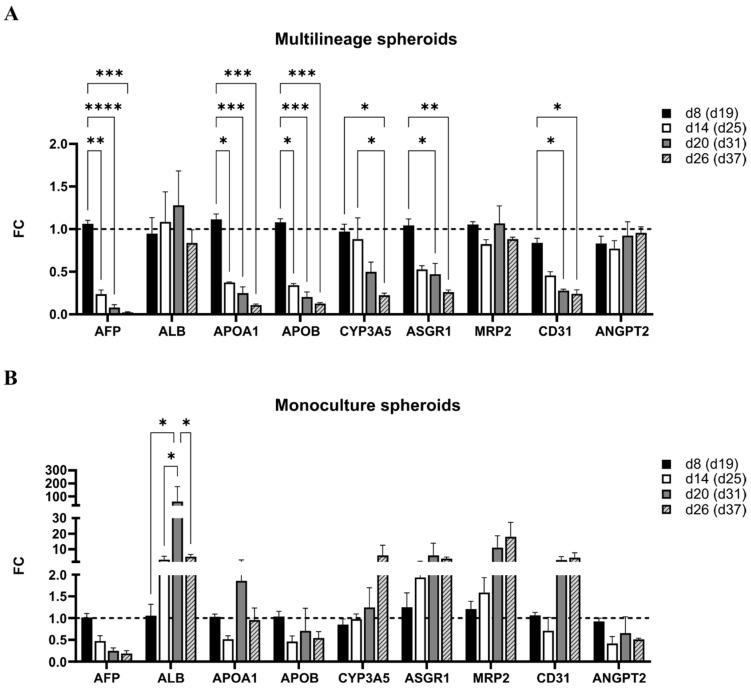
The expression of liver- and endothelial cell-related genes during spheroid culturing in (**A**) multilineage spheroids of iPSC-HLC+HUVEC+hASC and (**B**) iPSC-HLC monoculture spheroids at time points d8, d14, d20, and d26, corresponding to hepatic differentiation days 19, 25, 31, and 37, respectively. Results are expressed as fold change (FC) relative to spheroid d8 (i.e., diff. d19) of each spheroid type (FC = 1 denoted with dashed line). Bars represent mean ±SEM (three separate experiments, at least two biological replicates of each time point). Two-way ANOVA and Tukey’s multiple comparison test were performed, *p*-value * <0.05, ** <0.01, *** <0.001, and **** <0.0001.

**Figure 8 cells-12-02368-f008:**
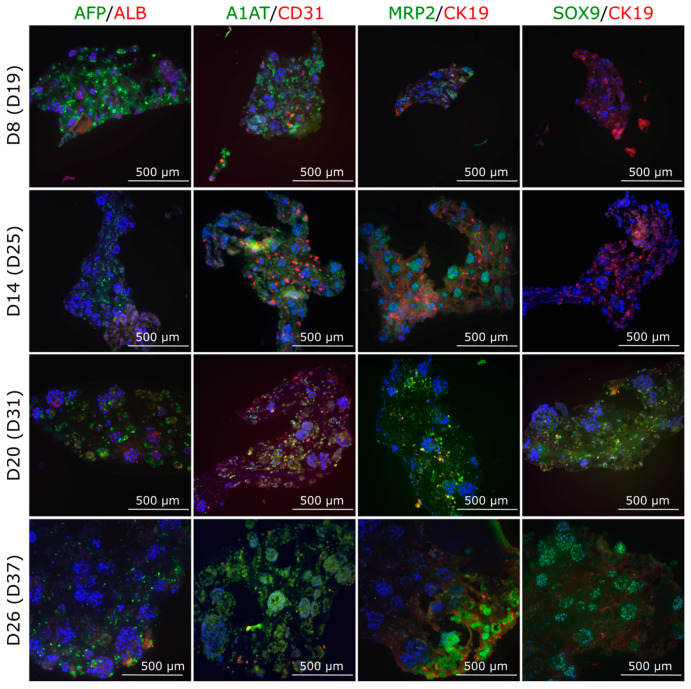
Expression of various liver markers along with endothelial cell marker CD31 in the multilineage spheroids composed of iPSC-HLCs, HUVECs, and hASCs. Spheroids were fixed, mounted in paraffin, cut in 4 μM sections, and stained at spheroid days 8, 14, 20, and 31, corresponding to differentiation days 19, 25, 31, and 37, respectively. Nuclei were stained with DAPI (blue). Scale bars 500 µM. The stained spheroid sections were imaged with an Olympus IX51 Fluorescence microscope (Olympus Corporation). Corresponding fluorescence intensity quantification for all markers can be found in Appendix A. ALB, albumin; AFP, alpha-fetoprotein; A1AT, alpha 1 antitrypsin; CD31, cluster of differentiation 31 (a marker for endothelial cells); MRP2, multidrug resistance-associated protein 2; and CK19, cytokeratin 19 (hepatocyte progenitor/cholangiocyte marker).

**Figure 9 cells-12-02368-f009:**
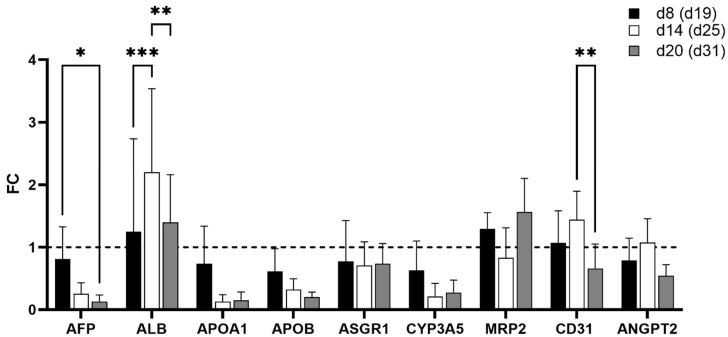
Gene expression levels in multilineage chips (AIM Biotech) with iPSC-HLCs, HUVECs, and hASCs (20:5:1) cultured for 8, 14, and 20 days on-chip, corresponding to differentiation days 19, 25, and 31, respectively. There were three chips at each time point in three separate experiments, the mean + SD of which is presented. Statistical significance was calculated with two-way ANOVA and Tukey’s multiple comparison test; *p*-value * <0.05, ** <0.01, and *** <0.001.

**Figure 10 cells-12-02368-f010:**
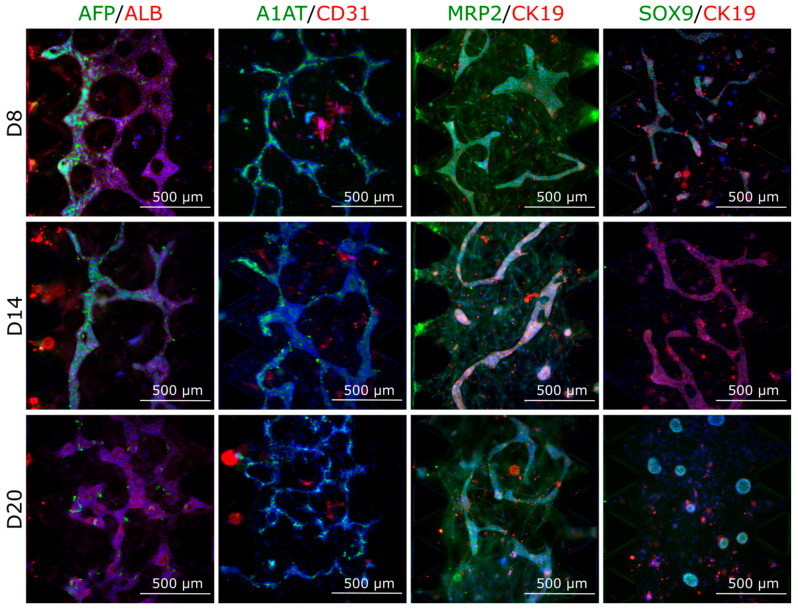
Expression of hepatic and endothelial marker proteins in the multilineage chips. The iPSC-HLCs were cultured with endothelial cells (HUVECs) and hASCs in ratios 20:5:1 on microfluidic chips (AIM Biotech) for 8, 14, or 20 days, corresponding to hepatocyte differentiation days 19, 25, and 31, respectively. The chips were imaged with an Olympus IX51 Fluorescence microscope (Olympus Corporation), nuclei were stained with DAPI (blue). Scale bars 1000 µm. Corresponding fluorescence intensity quantification for all markers can be found in Appendix A. ALB, albumin; A1AT, alpha 1 antitrypsin (mature hepatocyte markers); AFP, alpha-fetoprotein; CD31, cluster of differentiation 31 (a marker for endothelial cells, i.e., HUVECs); CK19, cytokeratin 19 (hepatocyte progenitor/cholangiocyte markers); and MRP2, multidrug resistance protein 2.

**Figure 11 cells-12-02368-f011:**
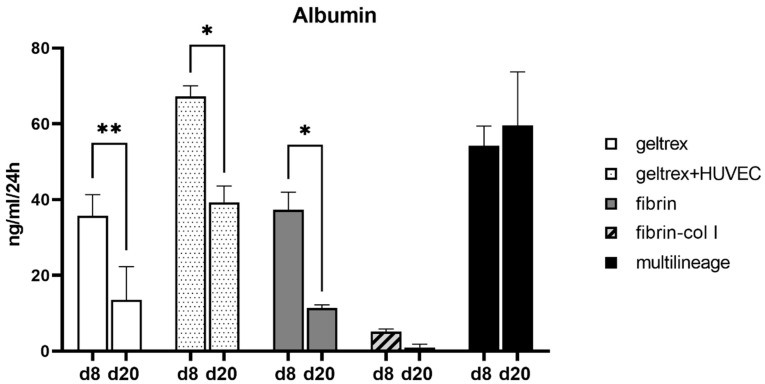
Production of albumin in the chip cultures stays constant in multilineage chip cultures from d8 to d20 on-chip. The iPSC-HLCs were cultured as monocultures or in co-culture with HUVECs in Geltrex, as monocultures in fibrin, or as multilineage cultures of iPSC-HLCs, HUVECs, and hASCs in fibrin. Bars represent mean ±SD. Statistical significance was calculated with 2w ANOVA and Šidák’s (d8 vs. d20) multiple comparison tests, *p*-value * <0.05, and ** <0.01.

**Table 1 cells-12-02368-t001:** Details of the biomaterial tests and co-culture experiments with HUVECs performed with iPSC-HLCs on the microfluidic chip (AIM Biotech; each chip has three gel channels each flanked by two medium channels). Seeding day refers to the day of hepatocyte differentiation at the day of chip loading.

Exp	Cell Type	No of Chips	Seeding Day	Biomaterial	Time Points	Medium
1	iPSC-HLC	5	d10	Geltrex (3:2)	d8, d20	HCM
2	iPSC-HLC	4	d10	Geltrex (2:1)	d8, d20	HCM
3	iPSC-HLC	5	d10	fibrin	d8 (term)	HCM
4	iPSC-HLC	9	d11	fibrin	d8, d20	HCM+apr
5	iPSC-HLC	6	d11	col I	d8, d20	HCM
6	iPSC-HLC	10	d11	fibrin-col I	d8, d20	HCM+apr
7	iPSC-HLC+ HUVECs in media channels	2	d10	Geltrex 3:2	d8, d20	HCM:EGM
8	iPSC-HLC+ HUVECs in media channels	2	d10	Geltrex 2:1	d8, d20	HCM:EGM
9	iPSC-HLC + GFP- HUVECs mixed in gel channel	6	d10	Geltrex 2:1	d8, d20	HCM:EGM

iPSC, induced pluripotent stem cell; HLC, hepatocyte-like cell; col I, collagen I; HCM, hepatocyte culture medium supplemented with 25 ng/mL HGF and 20 ng/mL OSM; term, experiment terminated; HUVEC, human umbilical cord endothelial cell; EGM, endothelial growth medium; and apr, aprotinin (5 μM).

## Data Availability

The datasets generated during and/or analysed during the current study are available from the corresponding author on reasonable request.

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
