# Peer review of "Improvements in Maturity and Stability of 3D iPSC-Derived Hepatocyte-like Cell Cultures"

_cells, 2023, doi:10.3390/cells12192368_

Round 1

Reviewer 1 Report (Previous Reviewer 4)

The article titled "Improvement in maturity and stability of 3D iPSC-derived hepatocyte-like cell cultures" establishes 3D cultures of iPSC-HLCs alone or co-culture with HUVECs and hASCs. These 3D in vitro liver systems can be used for various liver-related studies. This article has been modified according to the request of the reviewer.

Please make minor editing to the English language.

Author Response

Reviewer 2 Report (Previous Reviewer 3)

Authors have successfully generated a maturated liver-like structure with sustained protein expressions using 3D culture systems. The current work will contribute to the development of an in vitro system to assess the hepatotoxity in drug discovery. 

Minor concerns:

To underscore the importance of the current work, a description regarding the possible involvement in the development of an in vitro system to evaluate the hepatotoxity in drug discovery should be added in Abstract or Discussion. 

Author Response

Reviewer 3 Report (Previous Reviewer 2)

Thank you for addressing previous comments. No additional comments.

Best of luck in your research.

Author Response

Reviewer 4 Report (New Reviewer)

Suominen et al. have conducted research developing in vitro liver models using iPSC-derived hepatocyte-like cells. Using 3D culture and co-culturing platform, they showed that both 3D spheroid and on-chip culture enhance the expression of mature liver-marker genes and proteins compared to 2D cultures. However, there are limitations and concerns, as mentioned below, that the authors should address before the acceptance of this work.

1)    Please include a bright field of 2D culture of iPSC-derived hepatocytes to validate successful differentiation by showing a typical hepatocyte morphology (or at least hepatocyte-like morphology).

2)    There is another culture method besides 2D and spheroid for hepatocytes/iPSC-hepatocytes, which is “sandwich culture.” It is an important culture method the field used to overcome several shortcomings in the 2D culture. Please include a discussion to be comprehensive.

3)    A major concern is the lack of control in the experiment measuring mRNA levels of hepatocyte markers. There are commercially available primary human hepatocytes available. Please use them as the control to test the differentiation and expression of hepatocyte markers in iPSC-HLC, as well as albumin and urea production.

4)    Showing representative images of IHC is not a measurement of protein expression. Please at least include semi-quantitative measuring of fluorescent intensity of the staining of different targets in multiple fields of different spheroids/on-chip cultures.

5)    In Figure 4, the urea production measured in 2D cultured cells is flat with little variation throughout the time course, and at such a low range (ng/ml) indicates that the value measured could be below the limit of detection. Judging from the big variation in albumin concentration from 3D cultured vs. the little variation in 2D cultured, as well as the urea results, it is very likely that the measurements for both albumin (2D) and urea (2D and 3D) are below the limit of detection. To counter this point, please show the standard curves of these assays.

6)    Key evidence is missing to demonstrate that the cells are “hepatocyte-like” with no significant difference in albumin production, RNA expression markers are not compared to primary hepatocytes. One suggestion, since the authors are already using microfluidic channels, an experiment studying the drug metabolism function that could demonstrate the differentiation towards hepatocytes is a drug loading test, such as with acetaminophen. A difference in viability between 2D and 3D cultured cells should be seen as the drug concentration increases.

Round 2

Reviewer 4 Report (New Reviewer)

Most concerns have been addressed, and the authors made some good arguments by citing their previous work to address my question about PHH control. 

This manuscript is a resubmission of an earlier submission. The following is a list of the peer review reports and author responses from that submission.

Round 1

Reviewer 1 Report

This MS aims to optimise the 3D iPSC-derived liver models as multilineage spheroids in suspension or on microfluidic chips. It shows some preliminary data regarding the extension of the hepatocyte function. It compares iPSC-derived hepatocyte maturation/function in different cell culture types: 2D and 3D, in different co-cultures and without/with flow (on microfluidic chip). The claimed beneficial of the 3D co-cultures comes mainly from higher expression/secretion of albumin. The other read-outs are not clear, showing mainly a very heterogeneous system with a lot of debris. However, the presented read-outs, both from the immunofluorescence and PCR approaches, are not convincing and clearly showing that the protocols include a robust optimisation. While very expensive and extensive, the procedures in this MS do not show novel aspects and a clear progress in manufacturing iPSC-derived hepatocyte. The preliminary data here could be helpful for further optimisation, with better read-outs and controls.

English is good.

Reviewer 2 Report

The establishment of physiologically accurate and durable in vitro hepatic models is of high interest in toxicology and regenerative medicine. The manuscript by Suominen et al. provides an interesting overview and investigation of potential avenues for improvement of those highly sought-after models.

General comments:

The language tone utilized in the manuscript often sounds very colloquial, which could be more appropriate for a thesis or book chapter. Grammar and stylistic revisions to conform to journal manuscript stylistic formats are recommended.

The expression of certain markers analyzed for cell identity and cell maturity in immunohistochemistry have not been evaluated by qPCR. Given that the latter is generally a higher throughput method where more markers can be evaluated the authors should consider analyzing those transcripts as well. Specific examples are indicated in the section below.

The discussion feels excessively long and at times verbose. It should be more focused and shortened to be more concise, thus improving the flow and readability of the manuscript.

Specific comments:

Line 97: The exponent of cell density mentioned is apparently not formatted correctly.

Results section:

Figs 2, 4, 5A and 5B, 6 and 9 - The use of different time scales, different “days” for 2D culture and after 3D spheroid formation and the other complex formats can be confusing. The days following the re-formatting of the cultures could be omitted for simplicity.

The expression of CK19 is analyzed by Immunohistochemistry in figures 3, 7, and 8, and of A1AT in figures 3 and 7. Given that a number of other relevant markers have also been probed by qPCR in the preceding  figures 2, 5 A and B, 6 and 9, CK19 and A1AT should be included in those panels as well.

How do the authors reconcile the decline in ALB transcript expression after D20 in fig 2 with the increased and sustained protein expression observed in the ELISA assay in fig 4?

Line 450 The authors should elaborate on why they believe that the medium composition potentially affected ALB expression.

Section 2.10.2. This portion of the manuscript is very descriptive with little to no data accompanying it. It could be removed as it does not provide critical insights to subsequent experiments.

The language tone utilized in the manuscript often sounds very colloquial, which could be more appropriate for a thesis or book chapter. Grammar and stylistic revisions to conform to journal manuscript stylistic formats are recommended.

Reviewer 3 Report

In the current study, authors successfully generated a liver-like structure containing AFPlow/AlBhigh hepatocyte-like cells using 3D multilineage culture systems using iPSC-HLCs, HUVEC and hASCs. They also showed that a chip culture system has an advantage over a simple spheroid culture from the standpoint of sustained protein production.

Although the study clearly shows a technical advancement in hepatocyte differentiation of human iPSC cells, it seems rather difficult for readers to smoothly understand the novelty and the worth of the study due to unclear expressions. It is strongly recommended that Title and Abstract are changed so that readers can easily recognize the technical advancement in the differentiation of hiPSC into mature (or maturing) hepatocytes in vitro, which is distinct from the generation of immature liver buds that undergo maturation in vivo after transplantation (Takabe et al. Nature 499:481-484, 2013, and Ref 41).

The points of revision:

1)Since it is widely known that 3D culture has an advantage over 2D culture in inducing differentiations, the phrase "from 2D monocultures to 2 3D multilineage" in the Title should be omitted to avoid readers’ misunderstanding regarding the worth of the study.

Below is an example of a revised version of Title.

"Improvement in maturity and (protein expression) stability of human iPSC-derived hepatocytes by 3D multilineage chip cultures"

2)The sentence in Abstract "We saw the best functionality in iPSC-HLC monoculture spheroids whereas with the chip setting, the multilineage model performed the best." is rather confusing since the corresponding data are presented in the very last figure (Fig. 11). This sentence should be changed by referring more specific points.

Below is an example of a revised version.

"Although iPSC-HLC monoculture spheroids demonstrated higher expressions in hepatocyte-specific genes, 3D chip multilineage cultured cells surpassed them from the point of sustained protein expressions."

Reviewer 4 Report

Manuscript Review

The article entitled "iPSC-derived hepatocyte-like cells-from 2D monocultures to 3D multilineage cultures" established 3D culture of iPSC-HLCs alone or co-culture with HUVECs and hASCs. These 3D in vitro liver systems can be used to various liver-related studies. Below are my suggestions.

Keywords: I think there are too many keywords.

The section of "Materials and Methods" and "Results" should be simplified. I think the authors should carefully sort out the logical order of the results.

Results:

I think manuscript is lack of identification of iPS-induced hepatocyte-like cells.

In Figure 2 "Statistical significance was calculated with 2W ANOVA and p-values are from Dunnett’s multiple comparison test", I think One way ANOVA should be used.

In Figure 3, After spheroid sections staining, please mark which is AFP or ALB, MRP2 or CK19? Please give magnification in Fig. 3B and 3C.

In Figure 4B, the data of urea production of d26 in 3D culture system and d8 and d 26 in 2D culture system had no error bar, Is it because there is no repeat?